# Antitumor Effects of Ral-GTPases Downregulation in Glioblastoma

**DOI:** 10.3390/ijms23158199

**Published:** 2022-07-25

**Authors:** Tània Cemeli, Marta Guasch-Vallés, Marina Ribes-Santolaria, Eva Ibars, Raúl Navaridas, Xavier Dolcet, Neus Pedraza, Neus Colomina, Jordi Torres-Rosell, Francisco Ferrezuelo, Judit Herreros, Eloi Garí

**Affiliations:** 1Cell Cycle Group, Department of Basic Medical Sciences, Institut de Recerca Biomèdica de Lleida (IRBLLEIDA), University of Lleida, 25198 Lleida, Spain; tania.cemeli@udl.cat (T.C.); marta.guasch@udl.cat (M.G.-V.); mribes@irblleida.cat (M.R.-S.); eva.ibars@udl.cat (E.I.); neus.pedraza@udl.cat (N.P.); neus.colomina@udl.cat (N.C.); jordi.torres@udl.cat (J.T.-R.); francisco.ferrezuelo@udl.cat (F.F.); 2Oncopathology Group, Department of Basic Medical Sciences, Institut de Recerca Biomèdica de Lleida (IRBLLEIDA), University of Lleida, 25198 Lleida, Spain; raul.navaridas@udl.cat (R.N.); xavi.dolcet@udl.cat (X.D.); 3Calcium Signaling Group, Department of Basic Medical Sciences, Institut de Recerca Biomèdica de Lleida (IRBLLEIDA), University of Lleida, 25198 Lleida, Spain; judit.herreros@udl.cat

**Keywords:** glioma, glioblastoma, Ral-GTPases, RalB, recurrence, therapy

## Abstract

Glioblastoma (GBM) is the most common tumor in the central nervous system in adults. This neoplasia shows a high capacity of growth and spreading to the surrounding brain tissue, hindering its complete surgical resection. Therefore, the finding of new antitumor therapies for GBM treatment is a priority. We have previously described that cyclin D1-CDK4 promotes GBM dissemination through the activation of the small GTPases RalA and RalB. In this paper, we show that RalB GTPase is upregulated in primary GBM cells. We found that the downregulation of Ral GTPases, mainly RalB, prevents the proliferation of primary GBM cells and triggers a senescence-like response. Moreover, downregulation of RalA and RalB reduces the viability of GBM cells growing as tumorspheres, suggesting a possible role of these GTPases in the survival of GBM stem cells. By using mouse subcutaneous xenografts, we have corroborated the role of RalB in GBM growth in vivo. Finally, we have observed that the knockdown of RalB also inhibits cell growth in temozolomide-resistant GBM cells. Overall, our work shows that GBM cells are especially sensitive to Ral-GTPase availability. Therefore, we propose that the inactivation of Ral-GTPases may be a reliable therapeutic approach to prevent GBM progression and recurrence.

## 1. Introduction

Gliomas are brain tumors of glial origin that cause high mortality and morbidity as a result of their location, being very aggressive and resistant to radiotherapy and chemotherapy [1,2]. Among gliomas, the most frequent type is the glioblastoma (grade IV astrocytoma) (GBM). GBMs have the ability to infiltrate the surrounding brain parenchyma, complicating complete surgical resection. However, GBMs do not metastasize out of the brain [3]. In the last years, there have been multiple advances in the understanding of the molecular pathogenesis of GBM, and a new classification based on transcriptomics and patients’ genetic profile has been developed [4]. Thus, GBMs are divided into four main groups: proneural, neural, classical, and mesenchymal. In addition, the isocytrate dehydrogenase gene (*IDH1*) status is relevant for GBM treatment. Ten percent of GBMs harbor a point mutation (R142H) in the *IDH1* sequence gene. These IDH-mutant GBM (mostly classified as proneural) are less aggressive and have a better prognosis [5]. This complements the traditional histopathological classification [6], and it is important to find new targets for effective and personalized treatments.

The American Association for Cancer Research defines cancerous stem cells (CSCs) as a subpopulation of tumor cells with self-renewal capabilities, which can lead to a heterogeneous population of cancerous cells that make up the tumor [7]. In the case of gliomas, glioma stem cells (GSCs) present a persistent self-renewal capacity with the potential to give rise to cells of several differentiated lineages (glial and neuronal) and with the ability to form a new tumor when they are transplanted into model animals [8,9]. Moreover, GSCs transit into different microstates, showing a high adaptability to changes in the tumor environment that provide a means of therapeutic resistance [10]. GSCs present in the brain parenchyma after resection of the tumor most likely account for the recurrences.

Standard treatment for GBM involves surgical resection, followed by chemo- and radiotherapy. GBM chemotherapy entails the use of the alkylating agent temozolomide (TMZ), which produces DNA damage. Regarding this treatment, it is important to determine the methylation status of the O-6-methylguanine-DNA methyltransferase (*MGMT*) gene. This protein is capable of repairing DNA damages produced by alkylating agents [11]. Hence, tumors with silenced MGMT are sensitive to TMZ because they are not able to repair damaged DNA. These patients will have a better response to the treatment and consequently a better prognosis [2].

Ral-GTPases act as molecular switchers, presenting two different conformational states: an active state bound to GTP and an inactive state bound to GDP [12]. Ral-GTPases are involved in the control of different cellular processes such as exocytosis, cell migration, and cytokinesis [13,14,15]. Ral-GTPases are effectors of downstream Ras signaling and play an important role in Ras-driven tumorigenesis in human cells [16]. Classically, different studies in human cells have shown that RalA is important for tumor cell growth, whereas RalB is important for tumor cell survival and invasiveness [17]. The interaction of the active Ral GTPases with their downstream effectors, mainly Ral-BP1, Sec5, and Exo84, accounts for most of the Ral functions. For instance, RalA requires association with its Ral-BP1 and Exo84 effectors to induce anchorage-independent growth in colorectal cancer cells [18], and RalB effector Sec5 recruits and activates the TBK1 kinase, restricting the apoptotic response and promoting survival of different human cell lines [19].

Upregulated expression and activity of the Ral GTPases is frequently observed in almost all cancer types [17]. However, little is known about the role of Ral-GTPases in human malignant gliomas. The importance of geranylgeranyltransferase I (GGTase-I) in cancer progression and metastasis through membrane-targeting of Ral GTPases has been highlighted [20]. Other works have showed that overexpression of Ral-BP1 is associated with glioma grade and poor survival [21] and that Ral-BP1 knockdown reduces invasiveness, increases chemosensitivity to TMZ, and enhances the autophagy flux in these cells [22,23]. In a previous study, we showed that Ral-GTPase activity can contribute to GBM dissemination [24]. These data suggest that the inhibition of Ral GTPases may be therapeutically relevant in GBM. In this work, we show the antitumor effects of Ral-GTPases downregulation in GBM cells in vitro and in vivo.

## 2. Results

### 2.1. Expression of Ral-GTPases in Primary Human Glioblastoma Cells

First, we have analyzed the expression levels of RalA and RalB proteins in primary cultures obtained from human GBM and grade II astrocytoma biopsies (Figure 1A; Appendix A). Most of the GBM samples showed increased RalB levels compared to the low-grade astrocytoma samples. The expression levels of RalB in primary GBM cells were similar to the ones observed in the GBM cell line U251-MG (Figure 1a). By specifically pulling down RalB using beads containing the Ral-BP1 effector, we determined the levels of the active form of RalB (RalB-GTP) in two samples with different total RalB amounts. The result suggests that the increase in RalB amount implied an augmentation of the GTPase activity (Figure 1B; Appendix A).

A predominant expression of RalB in GBM samples is consistent with the data included in the GlioVis database (TCGA LGG_GBM dataset) and showed that transcriptional expression of RalB clearly increases in GBM samples in comparison with astrocytoma (Figure 1C). Moreover, among the GBMs, a lower expression of RalB is detected in the proneural subtype, which is less aggressive (Appendix A). Furthermore, the transcriptional expression of RalA also augments in GBMs versus astrocytoma, although to a lesser extent. In accordance with this, we observed RalA protein level increases in 4 out of 10 primary GBM samples, compared to astrocytoma samples (Figure 1A).

### 2.2. Relevance of Ral-GTPases in Glioblastoma Growth

To test the importance of Ral-GTPases on GBM growth, RalA and RalB GTPases were downregulated by using interference RNA in three different primary GBM cells that represent different behaviors regarding Ral GTPases expression: GBM65, GBM6, and GBM41. All of them were *IDH1* wild type [24]. Growth and viability of knockdown cells were determined by counting total and dead (trypan-blue-positive) cells. Downregulation of Ral-GTPases reduced growth without affecting cell viability (Figure 2a; Appendix A), suggesting an effect on cell proliferation. The double RalA and RalB knockdown showed the same level of growth reduction and viability as single knockdowns. Thus, the inhibition of only one of the Ral-GTPases may be enough to reduce cell growth in primary GBM cells. We have also observed that RalB downregulation produced morphological changes, larger size, and flattened cells in GBM primary cells (Figure 2b; Appendix A).

The growth reduction in the absence of RalA and/or RalB was also observed in soft agar colony assay in the U251-MG cell line and GBM cell cultures infected with scramble or interference RNA against RalA or RalB (Figure 3). Fifteen days after seeding, cell colonies were counted by MTT staining. Most knockdown cells did not produce colonies (Figure 3a), with a reduction of over 80% in colony formation (Figure 3b) compared to scramble shRNA cells.

### 2.3. Ral Downregulation Promotes a Senescence-like State in Primary Glioblastoma Cells

Our data showed that downregulation of Ral-GTPases reduces cell proliferation without affecting cell viability, which, together with the morphological changes observed, suggested a senescence-like response. To test senescence features, β-galactosidase assay was performed in primary GBM cells. Five days after the downregulation of Ral-GTPases, fixed cells were incubated in the presence of X-gal solution, and blue cells were counted (Figure 4a).

Knockdown of RalB (and RalA to a lesser extent) produced an increase in the number of X-gal-positive cells in comparison with controls (Figure 4b). To further evaluate the proliferation arrest, a BrdU incorporation assay was performed in primary GBM cultures. Five days post-infection with lentivirus driving the interference RNAs, cells were seeded and treated with BrdU for 12 h, and the accumulation of BrdU was tested by immunofluorescence (IF). We observed that the downregulation of RalB significantly decreased the number of BrdU-positive cells in all primary cultures compared to controls (Figure 4c; Appendix A). However, RalA downregulation only showed a significant decrease in GBM6 cells. Interestingly, RalB knockdown had a stronger effect than RalA in the senescence and BrdU assays.

Finally, in the same growth conditions, we measured the presence of cleaved caspase-3 in RalB knockdown cells (Appendix A). We have not detected caspase 3 cleavage, suggesting that the effect of RalB downregulation in primary glioblastoma cells does not trigger an apoptotic response.

### 2.4. RalB-Knockdown Induces a Senescent-like Response in Primary Glioblastoma Cells through Non-Canonical Mechanisms

Consistent with a proliferation arrest, senescent cells are often characterized by alteration in the p53-p21 and p16-Rb1 pathways [25,26]. The accumulation of these cell cycle inhibitors has been described in different glioma senescence models [27]. In addition, it has been shown that the downregulation of RalB induces the p53-p21 proliferation arrest pathway in diverse human tumor cell lines [28]. Therefore, we have investigated the significance of those pathways in primary GBM cells after RalB-knockdown. In GBM6 and GBM41, we have not detected the expression of p16 as previously described [24], nor p53 and p21, indicating the irrelevance of p53-p21 and p16-Rb1 pathways (Figure 5a: Appendix A). However, we have observed a significant accumulation of retinoblastoma protein Rb1, together with slight differences in the band mobility pattern of this protein in gels with a low percentage of acrylamide, after RalB-downregulation in GBM6 and GBM41 cells (Figure 5a,c). Upon senescence induction, hypo-phosphorylated Rb1 can be accumulated due to the p21 and p16-dependent inhibition of cyclin D-Cdk4/6 complexes [26]. Then, we checked the phosphorylation level of Rb1 after RalB downregulation in primary GBM cells. We observed the same levels of phosphorylation at Ser249/Thr252 at the N-terminus of Rb1 and at Ser780 at the C-terminus after RalB downregulation in control and shRalB cells (Appendix A). Thus, RalB downregulation would produce an accumulation of hypo-phosphorylated Rb1 in primary GBM cells. In addition, we have checked the levels of cyclin D1 and Cdk4, the most upstream Rb1 inhibitors. However, the levels of cyclin D1 and Cdk4 did not decrease after RalB downregulation in primary GBM cells (Appendix A). Therefore, additional mechanisms may explain the observed changes in Rb1.

The GBM65 cells showed a different behavior, expressing p16 and exhibiting a significant increase of p53 after RalB knockdown (Figure 5a,b). However, GBM65 cells did not express p21 (Appendix A) or Rb1 [24] (Figure 5a). Therefore, for GBM65 cells, the p53-p21 and p16-Rb1 pathways cannot explain the senescent phenotype either. Finally, the Cdk-inhibitor p27 (Kip1) has also been involved in senescent responses of GBM cells [29]. Even though we have detected the expression of p27 in primary GBM65 and GBM41, those cells did not show a consistent increment of p27 after RalB downregulation (Figure 5a,d). The overall data suggest that the arrest observed in primary GBM cells after RalB knockdown did not proceed from canonical pathways. Moreover, we have observed by cellular DNA content analysis that these primary cells did not show an efficient G1 arrest: a significant accumulation of G1 cells was not observed after RalB downregulation (Appendix A). It seems that the cell cycle could be arrested in the different phases after RalB downregulation in primary GBM cells.

It has been proposed that the DNA damage induced by different stimuli is a key upstream event in different pathways that promotes growth arrest and cell senescence [30]. To evaluate DNA damage in our cells, we have analyzed by immunofluorescence the levels of phospho-serine 139 H2A.X (γ-H2A.X) as a marker of double strand breaks. GBM65 cells showed a significant increment of nuclear γ-H2A.X foci when RalB was downregulated (Appendix A). This result opens the possibility that DNA damage induced by RalB downregulation could favor a senescent response in GBM65 cells. In contrast, GBM6 cells already showed a high proportion of DNA damage in both control and shRalB samples (Appendix A). In agreement with this result, it has been described that gliomas can have an aberrant activation of DNA damage signaling [31].

Finally, it has been proposed that an irreversible senescence state requires a cell division arrest coupled to high mTOR/S6K pathway activity [32]. We have analyzed the levels of active (phosphorylated) p70S6K kinase in the primary GBM cells after RalB knockdown. RalB-deficient cells accumulated phosphorylated p70S6K kinase (Figure 5a,e), indicating that the mTOR/S6K pathway was active in those GBM-arrested cells.

### 2.5. Ral-GTPases as Therapeutic Targets to Treat Glioblastoma

Several studies have proposed that GSCs may account for the initiation, progression, and recurrence of GBM [33,34,35,36]. Primary GBMs were cultured as tumorspheres (see methods), which are conditions favoring tumor stem cell proliferation. Whereas control (scramble) cells showed well-formed tumorspheres, single or double knockdowns of Ral-GTPases showed a huge reduction in the number of tumorspheres (Figure 6a,b). Moreover, in the case of Ral knockdowns, most of the cells in the sphere culture were positive for trypan-blue staining (Figure 6a,b). This result suggests that downregulation of Ral-GTPases reduces the viability of GBM cells growing in tumorsphere conditions.

Having observed the proliferation arrest and the senescent-like phenotype upon RalB knockdown, we decided to test the relevance of RalB downregulation in GBM cells in vivo. We performed mice subcutaneous xenograft by injecting U251-MG cells infected with either scramble shRNA or with shRNA against RalB (Figure 7a). To quantify tumor growth, we measured the volume of the tumor weekly (Figure 7b). We observed that the inoculum of control cells grew over time to form a tumor, while shRalB cells barely grew. In seven-week-old mice, proliferation was analyzed by Ki67 staining, and we observed that xenografted tumors from RalB-downregulated cells showed a significant reduction of proliferation (Appendix A). These results suggest that inhibiting RalB activity may be beneficial to controlling GBM growth. To reinforce this issue, we have used the TCGA dataset of diffuse gliomas (GBM-LGG) from the GlioVis platform [37] to obtain data on patient survival depending on the RALA and RALB genetic status (Figure 7c,d). Interestingly, an increase in the gene copy number of RAL genes was associated with a reduction in GBM patient survival. In contrast, the heterozygosis loss of RALB improved the long-term survival of glioma patients (Figure 7c).

Finally, we tested the role of RalB in TMZ-resistant cells. For this aim, we used GBM65 cells that express the MGMT protein and are consequently TMZ resistant and compared them with GBM6 cells, which are TMZ sensitive due to MGMT promoter methylation [24]. Both GBM65 and GBM6 cells were infected with lentivirus driving scramble or interference RNA against RalB and then treated with or without TMZ. RalB downregulation hindered proliferation of both primary GBM cultures, independently of the MGMT phenotype (Figure 8). Hence, inhibition of RalB activity may be a potential new therapy for GBM, specifically in those cases wherein tumors are insensitive to TMZ.

## 3. Discussion

Our data demonstrate that both RalA and RalB are required for the growth of primary GBM cells. The downregulation of only one of the Ral-GTPases was sufficient to reduce proliferation of primary GBM cells. In fact, the simultaneous knockdown of both Ral-GTPases produced a similar effect to knocking down only one. These findings suggest that RalA and RalB have overlapping functions in the control of the growth of GBM cells. Similarly, both Ral paralogs have overlapping effects in lung and melanoma cancer cells [38]. By contrast, other works have demonstrated that RalA is mainly involved in the control of anchorage-independent growth, while RalB regulates survival and invasion [39,40,41,42]. Moreover, our results indicate that the proliferation of GBM cells is particularly sensitive to Ral availability. Note that the knock-out of either RalA or RalB does not affect the growth of mouse embryonic fibroblasts (MEFs) [38], and slightly reduces the growth of some cancer cells, such as bladder tumor cells, when growing in low-serum conditions [41]. In MEFs and bladder cancer cells, the downregulation of both RalA and RalB activities is required to significantly reduce growth.

Ras-driven tumorigenesis can explain the increment of the Ral GTPases expression and activity in different cancer types [17]. In GBM cells, Ras mutations are not frequent; nevertheless, the Ras pathway is often activated by the overexpression of tyrosine-kinase receptors such as EGFR, which is common in this tumor [43]. Thus, Ras pathway hyperactivation could explain the upregulation of RalB (and to a lesser extent of RalA) in GBM cells. However, this is not conclusive, as it has been described that Ral-GTPase activation does not correlate with Ras status in melanoma and bladder tumors [44,45].

Ral-GTPases have multiple downstream effectors such as Ral-BP1 and different exocyst components. Interestingly, the overexpression of Ral-BP1 has been observed in GBM, associated with high tumor grade and poor survival [21,46]. Moreover, Ral-BP1 knockdown suppresses invasiveness, induces proliferation arrest, and increases chemosensitivity to TMZ [22,23]. These effects are similar to those induced by RalB-deficiency in primary GBM cells, thus reinforcing the relevance of the Ral pathway in GBM growth and suggesting that Ral-BP1 could be one of the main downstream effectors of Ral in GBM.

Although it was described that RalB inhibits apoptosis and promotes cell survival [19], we showed that the knockdown of RalB in primary GBM cells does not affect cell viability but decreases proliferation by inducing a senescence-like response. In agreement with our results, Tecleab et al. described that downregulation of RalB promotes growth arrest but not apoptosis in lung-cancer cell lines [28]. These authors showed that a reduction in RalB levels leads to the phosphorylation and stabilization of p53, which induces the expression of p21. Classically, the proliferation decline in the senescent response is associated with the activation of the p53-p21 pathway promoted by DNA damage and/or the accumulation of the Cdk4-inhibitor p16 preventing retinoblastoma (Rb1) inactivation [47]. However, our results in primary GBM cells indicate that RalB downregulation should induce a senescent-like response independently of the canonical pathways p53-p21 and p16-Rb1. Moreover, we cannot explain the senescent phenotype of RalB knockdown in GBM cells by the elevation of p27 either [29] [48]. The different primary GBM cells showed a loss of expression of several key proteins: p21, p16, and p53 in GBM41 cells; p21, p16, p27, and p53 in GBM6 cells; and p21 and Rb1 in GBM65 cells. In agreement with our data, Brennan et al. (2013) showed that GBM tumors frequently bear mutation in the *TP53* and *CDKN2A*/p16 loci and less frequently in the *RB1* locus [43]. Moreover, in previous work, we have already described that GBM6 cells were deficient for p53 and p16, whereas GBM65 cells did not express Rb1 [24]. We consider that the differences observed in our work (also for p21 and 27) reflect the genetic variability among GBM tumors, and it is difficult to find a common pathway involved in the senescence-like phenotype after RalB downregulation in GBM.

Our data suggest that there are different mechanisms promoting the senescence-like response due to RalB downregulation in GBM. In GBM65 cells, RalB downregulation promoted DNA damage and accumulation of p53. Even though the role of p53 has extensively been related with p21 upregulation, it has been described that p53 can trigger hepatocyte premature senescence through a p21-independent pathway [49]. This opens the possibility that p53 could also induce a senescence phenotype in GBM cells in the absence of p21 expression. Another possible mechanism could be related to the accumulation of Rb1 in GBM6 and GBM41 cells. Since we have not observed an increase of phosphorylated Rb1 forms after RalB downregulation, we assume the hypo-phosphorylated forms of this protein accumulate. The cell division arrest in senescence is associated with the accumulation of hypo-phosphorylated Rb1 through the p21 and p16 inhibition of cyclin D1-Cdk4 activity [26]. However, we have not detected either upregulation of p21, p27, and p16 or downregulation of cyclin D1 and Cdk4. Hence, we may explain the Rb1 accumulation through alternative pathways. Regarding this possibility, the induction of oncogenic senescence by mechanisms totally independent of p53-p21 and p16-Rb1 pathways has been described [50].

We cannot unquestionably conclude that downregulation of RalB in primary GBM induces a proper senescence arrest. However, cells showed some hallmarks associated with cell senescence, proliferation arrest, cell morphology changes, and ß-galactosidase activity. Moreover, cells continued to be viable, and we did not detect apoptotic responses. In addition, the accumulation of active p70S6K kinase seems to discard the possibility of a quiescent, transitory, arrest [32]. Some aspects remain elusive, such as the fact that GBM cultures did not show a G1 arrest after downregulation of RalB. This fact is not conclusive, however, as there are different examples demonstrating that the accumulation of G2 phase cells during senescence is also possible [26].

In this work, we propose that the inhibition of Ral-GTPase activity may be suitable for therapeutic interventions in GBM. The knockdown of RalB in GBM cells hinders the growth of a tumor mass in mouse-xenograft experiments. Consistent with this result, the heterozygosity loss of RalB implies a survival benefit for glioma patients. In addition, RalB downregulation prevents the growth of GBM cells in tumorspheres, suggesting that tumor stem cells are also affected. Interestingly, in tumorsphere growth conditions, we have observed cell death in most of the cells remaining as aggregates after RalB knockdown. This result supports the possibility that Ral activity could be required for cell survival of GSCs. This aspect is relevant in terms of the possibility of recurrences after GSCs have escaped from the tumor. Remarkably, we have shown that RalB knockdown is also efficient in reducing the growth of TMZ-resistant GBM cells, and therefore suitable for the treatment of MGMT-expressing GBMs. Unfortunately, we have been unable to confirm the efficiency of Ral-inhibitors BQU57 and RBC8 [51] in our experiments. Even though inhibitors from different sources and designs were tested, the results were not reproducible. This is likely due to the chemical instability of these molecules (D. Theodorescu, personal communication). Our results indicate that the inhibition of Ral-GTPases may be of great relevance in GBM treatment. Therefore, it will be crucial to develop novel potent Ral inhibitors that can be tested in clinics.

## 4. Materials and Methods

### 4.1. Cell Culture and Expression Vectors

GBM cell line U251-MG was obtained from CLS Cell Lines Service (Eppelheim, Germany). Primary GBM cell cultures were previously isolated from tumors from HUAV patients [52]. Briefly, tumor tissue was washed with phosphate buffer saline (PBS), cut into small pieces (2 mm^2^), and incubated (2 h, 37 °C under shaking) in PBS containing 155 U/mL of collagenase and 12 μg/mL of DNase-I. Samples were filtered through a 70 μm cell strainer, and the cell suspension was washed twice with PBS. Cells were plated in DMEM media containing 10% FBS. Media was changed every 2 days. Those primary cells were amplified (3–6 passages) from the original plate and aliquoted in liquid nitrogen. After re-seeding, stocked cells were used for experiments after 3–4 passages. In GBM6, 41, and 65 stocks, we never detected morphological or growth rate changes. Cell lines and primary cells were maintained at 37 °C in a 5% CO_2_ incubator, and grown in DMEM supplemented with 10% FBS, 100 μg/mL penicillin/streptomycin, and 2 mM glutamine. Cell lines were analyzed by immunofluorescence for specific glioblastoma markers such as GFAP and IDH1R132. Primary cell lines retained the expression of glial fibrillary acidic protein (GFAP) as reliable marker of astrocytic or glial cells [52] and were IDH wild type [24]. The methylation status of the MGMT promoter and the expression of MGMT were determined in GMB6 and GBM65 cells [24].

For lentivirus production, HEK293T cells were transfected with lentiviral expression vectors, envelope plasmid pVSV.G, and packaging plasmid pHR’82ΔR at a ratio of 2:1:1.

For RNA interference, the RalA MISSION shRNA TRCN0000004865 and RalB MISSION shRNA TRCN0000072957 cloned in a pLKO.1-puro were obtained from Sigma-aldrich: St. Louis, MO, USA.

### 4.2. Analysis of Expression by Immunoblotting

RalA and RalB protein amounts were analyzed by immunoblot. Protein extracts with SDS 2% were obtained from GBM and astrocytoma primary cultures. For immunoblot, protein samples were resolved by SDS-PAGE, transferred to PVDF membranes (Millipore, Burlington, MA, USA), and incubated with primary antibodies. Appropriate peroxidase-linked secondary antibodies (GE Healthcare, Chicago, IL, USA) were used and detected with the chemiluminescent HRP substrate Immobilon Western (Millipore, Burlington, MA, USA). Chemiluminescence was recorded with a ChemiDoc-MP imaging system (BioRad, Hercules, CA, USA). We have used mouse monoclonal antibodies anti-RalA clone 8 BD Pharmigen #610222, anti-β-actin clone C4 Millipore #MAB1501R, anti-p16 clone DSC50 Oncogene #NA29, anti-p21 Millipore #05-345, anti-p27 BD Biosciences #610242, anti-Rb1 BD-Pharmingen #5544136, anti-p53 Upstate #05-224, anti-Rb1 (pS780) BD-Pharmingen #558385, and anti-Rb1 (pS249/pT252) Santa Cruz #sc-377528. We have also used rabbit polyclonal anti-RalB Cell Signaling #3523, anti-Ccnd1 Millipore #ABE52, and anti-Cdk4 Santa Cruz #sc-260. For band quantification, we have used Image Lab software 4.0 from BioRad, Hercules, CA, USA.

### 4.3. Ral Pull-Down Assay

The Ral activation was analyzed by measuring the GTP-bound form of Ral. The assays were performed by using Ral-BP1 agarose (Upstate, cat# 14-415) according to the manufacturer’s instructions. GBM cell lysates were obtained from one 100 mm plate (1 × 10^6^ cells). The lysis buffer used was 50 mM Tris pH 7.5, 200 mM NaCl, 2.5 mM MgCl_2_, 2.5 mM DTT, 1% Triton X-100, and protease and phosphatase inhibitors. Cell lysate (0.6 mL) was incubated with 10μg of Ral-BP1 beads for 30 min at 4 °C, and after several washes, agarose beads were resuspended in 2x Laemmli buffer. Samples were separated by SDS-PAGE, transferred to PVDF membranes, and immunoblotted.

### 4.4. Proliferation and Viability Assays

For proliferation and viability assays, 15,000 cells per well were seeded on a 24-well plate in triplicate. After 24, 48, and 72 h of treatment, cells were trypsinized and counted in Neubauer’s chamber. For viability assays, 0.2% trypan blue was added before counting. In Neubauer’s chamber, we have counted the cells present in the 4 squares of 1 mm^2^, and the cell concentration (cells/mL) is calculated according to the manufacture’s formula: average number of cells in the four squares × dilution factor × 10^4^. Dilution factor is usually 2 (1:1 dilution with trypan blue). Dead cells were trypan blue positives. In Appendix A, the ratio of live cells was counted as (live cells)/(dead cells + live cells).

Proliferation and viability were also determined using the MTT assay. Cells were incubated at 37 °C and 5% CO_2_ for 1 h with MTT at 1 mg/mL in darkness. Then, media was removed, and DMSO was added to the wells to dissolve formazan crystals produced by living cells. Absorbance was determined at 595 nm wavelength.

### 4.5. Clonogenic Assays

For non-adherent conditions, cells were resuspended at a concentration of 3 × 10^3^ cells per ml in 0.3% agar diluted in culture medium. One ml of cell suspension (3 × 10^3^ cells) was added to a 6-well plate that was previously covered with a 0.6% agar layer. The cell plate was incubated for 15 days at 37 °C with 5% CO_2_ in a humid chamber. After this time, cells were treated with MTT for 1 h at 37 °C and 5% CO_2_ in a humid chamber in darkness. The number of colonies was counted using the digital image-processing program Image J.

### 4.6. Senescence-Associated β-Galactosidase Assay

For the analysis of SA-β-gal, cells were incubated for 10 min in PBS with 1 mM MgCl_2_, followed by incubation in X-gal solution—20 μg/mL X-gal (Sigma-Aldrich: St. Louis, MO, USA), 5 mM K_3_Fe(CN)_6_, 5 mM K_4_Fe(CN)_6_, and 2 mM MgCl_2_ in PBS at pH = 6. This incubation was carried out for 4 h (cell lines) or overnight (primary cell cultures). Finally, cells were fixed with 0.5% glutaraldehyde in PBS.

### 4.7. BrdU Incorporation

BrdU (Sigma) was added for 12 h into the media at a final concentration of 8 µg/mL. Then, cells were fixed for 15 min with 4% paraformaldehyde (PFA) at RT and permeabilized for 2 min with 0.2% Triton X-100. Next, samples were treated with 2M HCl for 30 min at 37 °C and neutralized with 0.1 M sodium tetraborate (pH 8.5) for 2 min. After neutralization, samples were blocked with 3% BSA for 1 h, washed, and processed with the antibodies. We have used the anti-BrdU rat monoclonal clone BU1/75 (ICR1) Bio-Rad #MCA2060 to detect positive cells under the microscope. Nuclei were stained with Hoechst 0.5 µg/mL (Sigma-Aldrich).

### 4.8. Immunofluorescence, Immunochemsitry and DNA Content Analysis

For immunofluorescence, cells were fixed with 4% PFA for 15 min at RT. Afterward, cells were permeabilized with 0.2% Triton-X100 for 3 min at RT and blocked with 3% BSA (Sigma) for 30 min. The primary antibody Anti-H2A-X (Ser139) clone JBW301 (Millipore #05-636) was diluted (1:200) in 0.3% BSA and incubated overnight at 4 °C. Next, the secondary antibody labeled with Alexa 488 (Molecular Probes, Eugene, OR, USA) at a 1:1000 dilution was added in 0.3% BSA in darkness at RT together with Hoechst (Sigma) to stain cell nuclei. Epifluorescence images were acquired in an inverted Olympus IX71 confocal microscope.

For IHC, sample sections of 3 μm were blocked for endogenous peroxidase and incubated with primary antibody Ki67 (Ready to Use (RTU), Dako, Glostrup, Denmark). The reaction was visualized with the EnVision FLEX Detection Kit (DAKO). Sections were counterstained with haematoxylin.

For determination of cellular DNA content, cells were trypsinized, fixed in 70% ethanol, and incubated for 30 min at 37 °C in 1× saline sodium citrate containing 50 µg/mL RNase A and 50 µg/mL propidium iodide prior to analysis by flow cytometry.

### 4.9. Tumorsphere Formation and Analysis

Cells in 2D cultures were trypsinized, centrifuged for 5 min at 180× *g*, washed with neurobasal media (GIBCO), and resuspended in tumorsphere media (50 mL neurobasal media + 1 mL B27 supplement + 500 µL Glutamax + 100 µL of penicillin/streptomycin + 100 µg FGF + 100 µg EGF), and 10^5^ cells were seeded in ultra-low attachment surface plates. Floating cells were transferred 24 h later into new ultra-low attachment surface plates for tumorspheres growth.

When they reach an appropriate size (five cells minimum), tumorspheres were concentrated by gravity in a 15 mL tube for 15 min. Then, tumorspheres were washed in PBS with trypan blue, centrifuged for 2 min at 5× *g*, resuspended in 500 µL PBS with a cut pipette tip, and moved into a 4-well plate. Images were analyzed with ImageJ, U.S. National Institutes of Health, Bethesda, MD, USA.

### 4.10. TCGA Data Analyses

To analyze the clinical relevance of our model, we obtained diagnosis, grading, survival, and gene status data of 669 patients with diffuse gliomas from The Cancer Genome Atlas Project (TCGA-GBMLGG) using the GlioVis platform (http://gliovis.bioinfo.cnio.es/, accessed on 5 July 2022).

### 4.11. Mouse Models

The procedure performed in this study followed the European Union Guidelines for the Care and Use of Laboratory Animals. The procedure is in accordance with the Law 5/1995 and the Decree RD53/2013, which regulate the use of animals for experimental and other scientific purposes (Catalan Government), and it was certified by the Ethics Committee on Animal Experimentation from the University of Lleida (CEEA 03-03/13).

Immunodeficient male SCID hr/hr mice (12-week-old; 20–25 g) were maintained in specific pathogen-free conditions, and U251 cells (8.75 × 10^5^) in 100 μL PBS + Matrigel (1:0.25) were subcutaneously injected in the flank. Tumors were allowed to grow for seven weeks. Mice were sacrificed by cervical dislocation, and the tumors were collected for macroscopic observation. Tumor volume was measured weekly by using a Vernier caliper and calculated according to the formula: tumor volume = (D × d^2^)/2, where D corresponds to the large diameter of the tumor, and d to the smaller one.

## 5. Conclusions

Primary GBM cells show increased amounts of RalB protein in comparison with primary astrocytoma cells. Downregulation of Ral GTPases RalB (and RalA in a lesser extend) decreases cell proliferation and induces a senescent-like response in primary GBM cells. RalB downregulation efficiently reduces tumor xenograft growth and diminishes proliferation in temozolomide-resistant GBM cells.

## Figures and Tables

**Figure 1 ijms-23-08199-f001:**
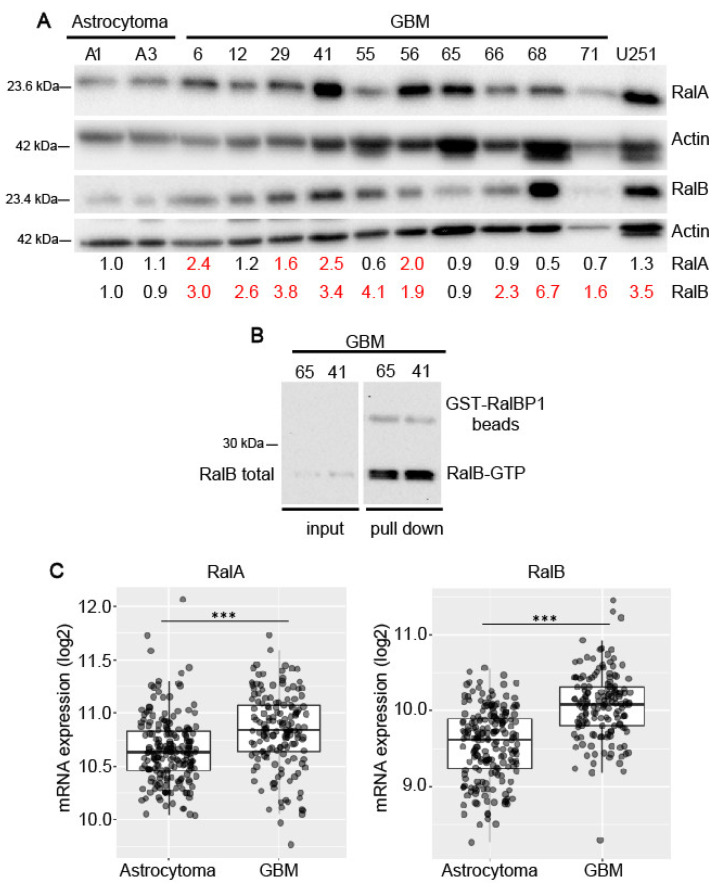
Expression of RalA and RalB in glioblastoma. (**A**) Immunoblot to detect the levels of RalA and RalB in different primary glioblastoma (GBM), in low-grade astrocytoma (A1, A3) primary cultures, and in the glioblastoma cell line U251-MG. β-actin was used as a loading control. Numbers below the panel A are the estimated levels of RalA and RalB relative to β-actin and refer to the astrocytoma sample A1. (**B**) RalB-GTP pull down in two different primary GBM cells. Active RalB-GTP was affinity purified with Ral BP-beads from cell lysates. RalB-GTP and total RalB were detected by immunoblot. Ral BP-beads were used as a loading control. (**C**) Analysis of RalA and RalB mRNA expression levels by RNAseq from GlioVis database (161 low-grade astrocytoma and 152 glioblastoma samples). Tukey’s Honest Significant Difference (HSD), *p* < 0.001 (***) (http://gliovis.bioinfo.cnio.es/, accessed on 5 July 2022).

**Figure 2 ijms-23-08199-f002:**
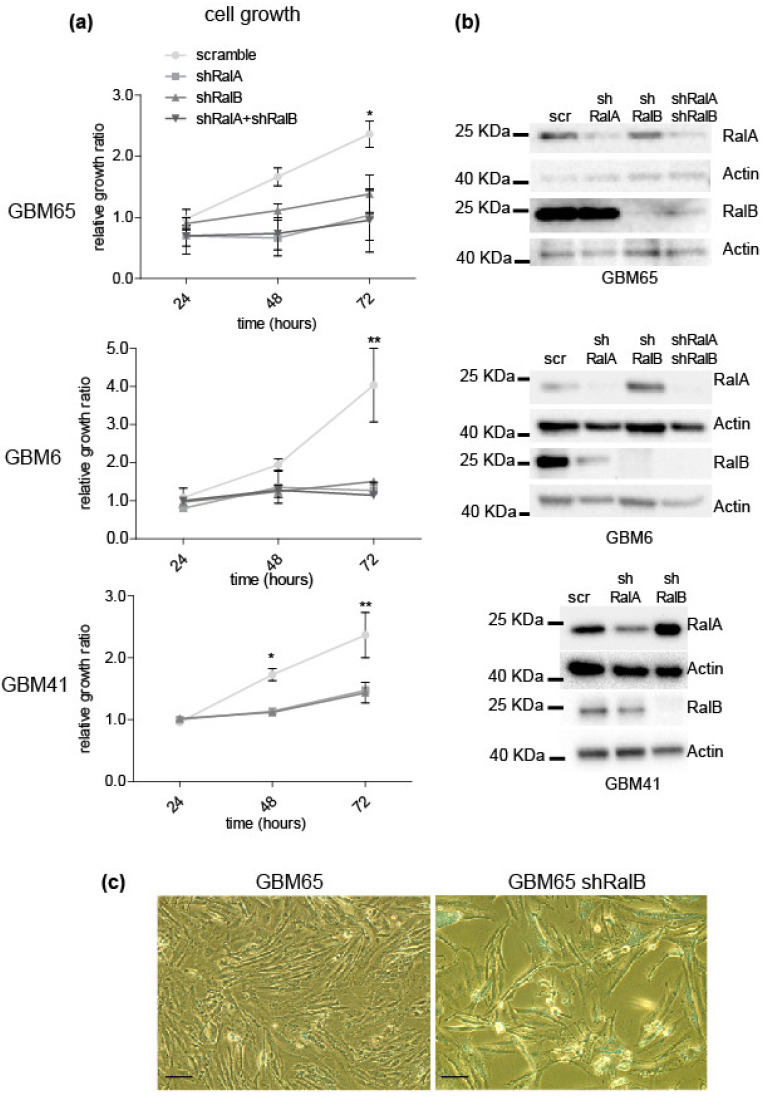
Knockdown of Ral GTPases promotes a reduction of cell growth without affecting cell viability. Primary GBM cells were infected with lentivirus, driving interference RNA against RalA (shRalA) or RalB (shRalB) or both. Scramble shRNA was used as a control. Three days after infection, cells were seeded and counted every 24 h for 3 days. (**a**) Graphics and bar diagrams representing growth of knockdown GBM cells. Data is represented as mean ± SEM (*n* = 3; three independent experiments). Scramble group was compared with the other three conditions by ANOVA and Tukey-HSD post-test (* *p* < 0.05; ** *p* < 0.01). (**b**) Western blot to determine the levels of RalA and B in primary GBM cells after knockdown. β-actin was used as a loading control. (**c**) Representative phase-contrast images of GBM65 cells were taken after 4 days of infection (50 μm bar).

**Figure 3 ijms-23-08199-f003:**
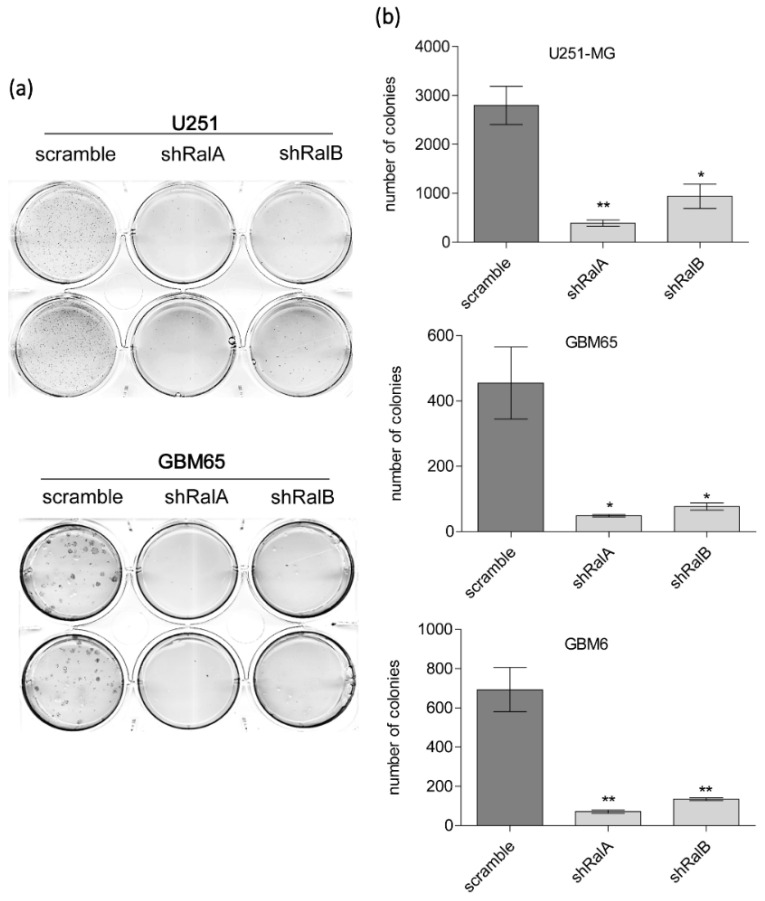
Knockdown of Ral GTPases reduces soft-agar colony growth. U251-MG cell line, GBM65, and GBM6 were infected with lentivirus-mediating interference RNA against RalA or RalB and seeded in a soft agar layer. After 15 days, the growing colonies were stained by MTT reaction, and the plates were scanned. (**a**) Representative images of U251-MG and GBM65 plates. (**b**) Diagram representing the number of colonies quantified with the ImageJ program. Values represent mean ± SEM (*n* = 4). Significance was calculated by ANOVA and Tukey-HSD post-test. Significance of scramble versus shRalA and shRalB groups is represented (* *p* ≤ 0.05; ** *p* ≤ 0.01).

**Figure 4 ijms-23-08199-f004:**
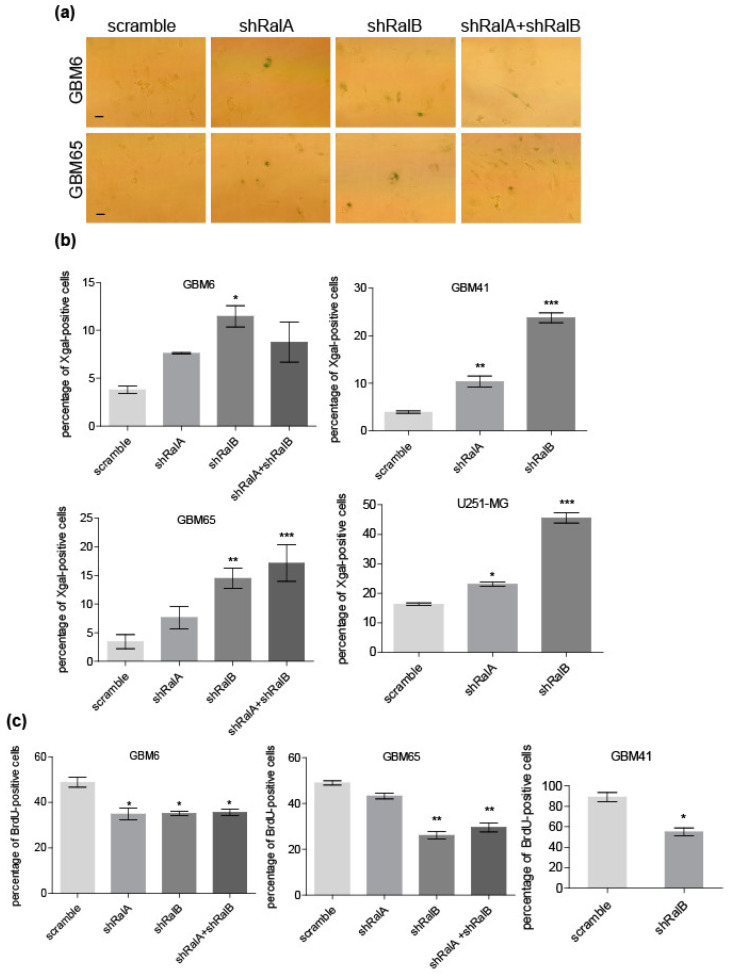
Downregulation of Ral GTPases promotes a senescence-like response in glioblastoma cells. Ral GTPases RalA and RalB were downregulated by RNA interference—shRalA and shRalB, respectively. Scramble shRNA was used as a control. GBM cells were infected by lentiviral vectors harboring the shRNAs. Five days after infection, cells were processed for senescence analyses. (**a**) Representative images of X-gal senescence assay. Blue cells indicate positive senescent cells (25 μm bar). (**b**) Quantification of senescence assay in (**a**) as a percentage of positive cells versus total cells. Values represent mean ± SEM. Significance was determined by ANOVA and Tukey post-test (*p* ≤ 0.05, *; *p* ≤ 0.01, **; *p* ≤ 0.001, ***). (**c**) Quantification of BrdU incorporation assay. Values represent the percentage of BrdU-positive nuclei. The number of total nuclei was determined by counting nuclei stained with Hoechst. The mean ± SD (*n* = 3) is shown. Significance was determined by ANOVA and Tukey post-test. Significance of scramble versus shRalA, shRalB, or shRalA + shRalB groups is represented (*p* ≤ 0.05, *; *p* ≤ 0.01, **).

**Figure 5 ijms-23-08199-f005:**
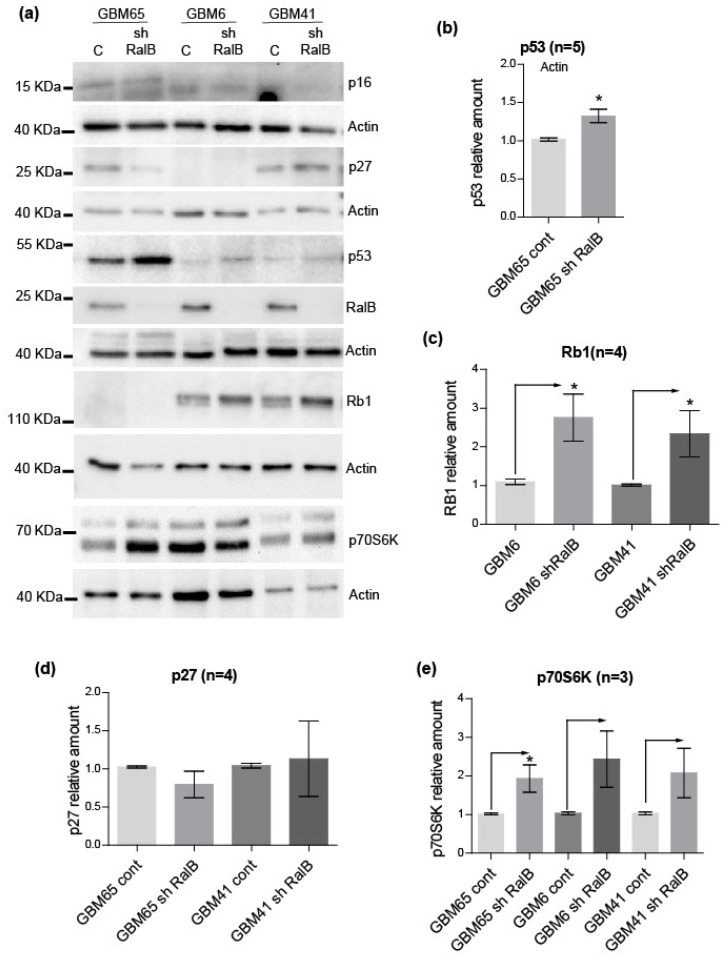
RalB knockdown induces a senescent-like response in primary glioblastoma cells through non- canonical mechanisms. Primary GBM cells growing in the same conditions as in Figure 4 were processed for immunoblot. (**a**) Panels showing the levels of the indicated proteins in primary GBM cells. Actin was used as a loading control. Quantification of the protein levels for p53 (**b**), Rb1 (**c**), p27 (**d**), and Phosphop70S6K (**e**). The number of independent experiments (*n*) is shown in the panels. The mean ± SEM is shown. Significance was determined by ANOVA or *t*-test (*p* ≤ 0.05, *).

**Figure 6 ijms-23-08199-f006:**
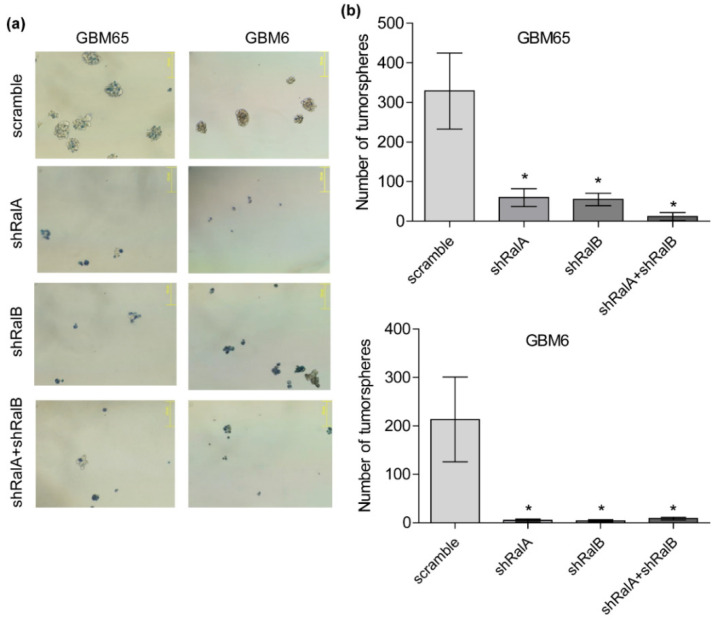
Ral GTPases downregulation inhibits the growth of GBM cells as tumorspheres. GBM65 and GBM6 were infected with lentivirus, mediating interference RNA against RalA and RalB or both, and cells were seeded under special conditions in order to form tumorspheres. After 48 h, tumorspheres were precipitated and stained with trypan blue. (**a**) Representative images of tumorspheres (150 μm bar). (**b**) Diagrams representing the number of spheres observed in (**a**). Spheres were counted using the ImageJ program. Values represent mean ± SEM (*n* = 3). Significance was calculated by ANOVA and Tukey-HSD post-test. Significance of scramble versus shRalA, shRalB, and shRalA + shRalB groups is represented (* *p* ≤ 0.05).

**Figure 7 ijms-23-08199-f007:**
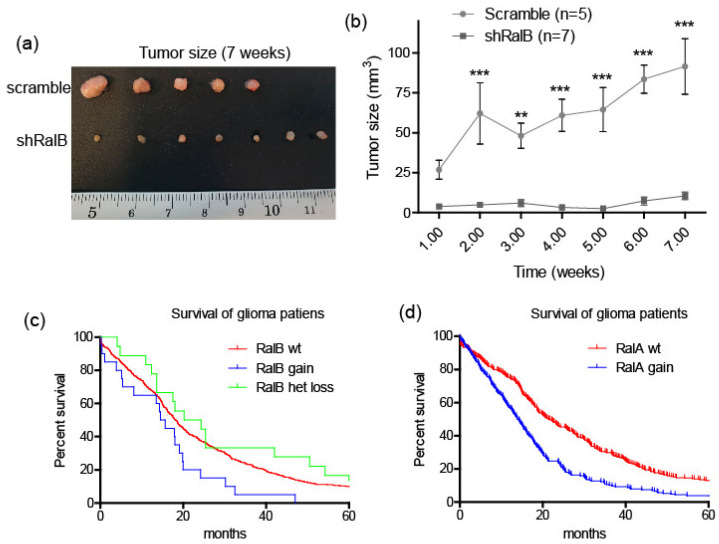
Ral GTPases downregulation prevents the growth of GBM cells in vivo. (**a**) Human U251-MG cells were infected with lentiviruses harboring scramble or shRalB. Infected cells were inoculated subcutaneously in immunodeficient SCID male mice. Tumor sizes were measured every week. Mice were euthanized seven weeks after injection and the tumors excised. (**b**) Quantification of tumor size at different weeks. Data is mean ± SEM. Significance was calculated by two-way ANOVA and Bonferroni-HSD post-test. Significance of scramble versus shRalB group is represented (** *p* ≤ 0.01; *** *p* ≤ 0.001). (**c**) Analysis of survival times (months) of glioma patients (TCGA cohort of diffuse gliomas, Gliovis) with a Kaplan–Meier plot comparing the effects of copy number increase of RalB (gain; *n* = 20), heterozygous loss of RalB (het loss; *n* = 18), and wild type RalB alleles (wt.; *n* = 600). The median survival in months was 15.1 (RalB gain), 18.1 (RalB wt.), and 22.25 (RalB het loss), with *p* = 0.048 (Mantel–Cox test). (**d**) Analysis of survival times (months) of glioma patients as in (**c**). RalA copy number increase (gain; *n* = 226) and wild type RalA alleles (wt.; *n* = 417). The median survival was 13.8 (RalA gain) and 21.55 (RalA wt.), with *p* < 0.0001 (Mantel–Cox test).

**Figure 8 ijms-23-08199-f008:**
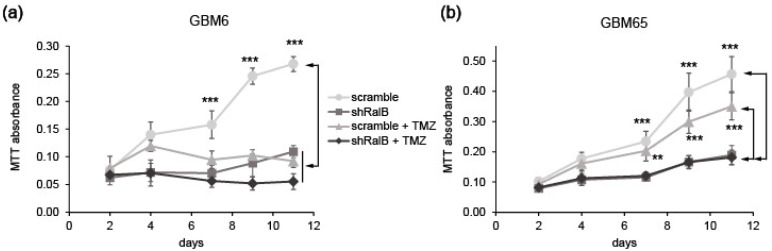
Effects of the treatment of primary glioblastoma cells with temozolomide (TMZ) and RalB downregulation. GBM6 (**a**) or GBM65 (**b**) cells infected with lentivirus driving scramble or shRalB were treated with TMZ (100 μM) or placebo. Growth was analyzed by MTT assay at different time points. Values represent mean ± SEM (*n* = 3). Significance was determined by ANOVA and Tukey-HSD post-test. Significance of scramble ± TMZ versus shRalB ± TMZ groups is represented (** *p* < 0.01; *** *p* < 0.001).

## Data Availability

Not applicable.

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
