# Peer review of "Antitumor Effects of Ral-GTPases Downregulation in Glioblastoma"

_ijms, 2022, doi:10.3390/ijms23158199_

Round 1

Reviewer 1 Report

The article entitle "Antitumor effects of Real-GTPases down regulation in glioblastoma" showed that the downregulation of RalA and RalB could be used as a therapeutic approach to prevent GBM progression and recurrence.

1.     Authors suggest an appropriate research design with a lot of results; although the methods are not adequately described. I think that is better to redraft the Material and Methods.

2.     The paragraph "Results" begins with the evaluation of the Ral expression, while in the materials and methods, the first paragraph concerns the evaluation of cell proliferation. I think it is better to organize the results according to M&M.

3.     The conclusions must be improved and supported by the results.

4.     Authors should specify what represents the numbers above and below the Figure 1A.

5.     In Figure 1C qPCR results are described, but in Materials and Methods the paragraph describing the qPCR method is not present. It should be added.

6.     In paragraph 2.2 you used three different primary GBM cells: GBM65, 116 GBM6, and GBM41. Why did you choose these cell lines? 

7.     Did you use hematocytometer to count total and dead cells? To calculate cell viability, did you use a formula? Authors should add the formula: 100*(live cells)/(dead cells + live cells).

8.     Why the authors did not evaluate the double RalA and RalB knockdown in GBM41 cell line (Figure 2A-B)?

9.     In paragraph 2.3. it is reported “the accumulation of BrdU was tested by immunofluorescence (IF)”. Where are the IF images? 

10.  In paragraph 2.4. the authors say that the expression of p16, p53, p21 and Rb1 is different in GBM6 and GBM65 cell lines. What does this different expression depend on?

Author Response

Comments and Suggestions for Authors R1

The article entitle "Antitumor effects of Real-GTPases down regulation in glioblastoma" showed that the downregulation of RalA and RalB could be used as a therapeutic approach to prevent GBM progression and recurrence.

  1. Authors suggest an appropriate research design with a lot of results; although the methods are not adequately described. I think that is better to redraft the Material and Methods.

      As requested by the reviewer, now we have redraft M&M. We have organized better according with the results (see below).

  1. The paragraph "Results" begins with the evaluation of the Ral expression, while in the materials and methods, the first paragraph concerns the evaluation of cell proliferation. I think it is better to organize the results according to M&M.

      We have organized M&M according the results. We have started results by analyzing the expression of Ral GTPases in different GBM primary cultures to later study the biological effects in some primary cultures. For that reason, we have maintained the results organization and redraft M&M.

  1. The conclusions must be improved and supported by the results.

      We have rewritten the conclusions: “Primary GBM cells show increased amounts of RalB protein in comparison with primary astrocytoma cells. Downregulation of Ral GTPases RalB (and RalA in a lesser extend) decreases cell proliferation and induces a senescent-like response in primary GBM cells. RalB downregulation efficiently reduces tumor growth in mice and also diminishes proliferation in temozolomide-resistant GBM cells.”

  1. Authors should specify what represents the numbers above and below the Figure 1A.

      Now, we have noted this in the figure 1 legend: “Numbers downstream the panel A are the estimated levels of RalA and RalB relative to b-actin and to referred with the Astrocytoma sample A1.”

  1. In Figure 1C qPCR results are described, but in Materials and Methods the paragraph describing the qPCR method is not present. It should be added.

We have not well explained the data in figure 1C in the text. The data (http://gliovis.bioinfo.cnio.es/) consist in mRNA levels of RalA and RalB (152 GBM and 161 Astrocytoma) obtained by RNAseq experiments. Now, we have added this information in the figure 1 legend.

  1. In paragraph 2.2 you used three different primary GBM cells: GBM65, 116 GBM6, and GBM41. Why did you choose these cell lines?

      From the 10 primary samples of GBM (Figure 1), GBM 6 and 65 are among the first we established in culture. Later, we also established GBM41. GBMs show a high variation in the genetic profile and accordingly we also found differences in the expression of Ral GTPases. GBM6 and 41 have high levels of both RalB and RalA, while GBM65 is one of the few GBM that does not have high levels of either RalB or RalA. It seemed to us that these three cultures represented the genetic variability inherent in GBMs. Now, we have commented this feature at the beginning in the results section 2.2.

  1. Did you use hematocytometer to count total and dead cells? To calculate cell viability, did you use a formula? Authors should add the formula: 100*(live cells)/(dead cells + live cells).

      We have used a Neubauer’s chamber counting the cells present in the 4 squares of 1 mm2 and the cell concentration (cells/ml) is calculated according to the manufacture’s formula: average number of cells in the four squares x dilution factor x 104. Dilution factor is usually 2 (1:1 dilution with trypan blue). Dead cells were trypan blue positives. In figure S2a, the ratio of live cells was counted as (live cells)/(dead cells + live cells). Now, we have added all this information in methods section 4.4. Proliferation and viability assays.

  1. Why the authors did not evaluate the double RalA and RalB knockdown in GBM41 cell line (Figure 2A-B)?

The establishment of the primary cultures was carried out immediately after receiving the biopsy sample and, after few passages, the yield in cell number was variable among the different samples. As cited above, from the 10 primary samples of GBM (Figure 1), GBM6 and GBM65 are among the first we established in culture. Later, we also established GBM41. In the first phase of experiments, in the case of GBM41, the availability of cells was lower and we considered more important do not increase the number of passages. As the results of GBM6 and GBM65 showed that the double knockdown (RalA RalB) was not different from the single knockdowns, we assumed the results obtained only with the single knockdowns to be adequate.

  1. In paragraph 2.3. it is reported “the accumulation of BrdU was tested by immunofluorescence (IF)”. Where are the IF images?

      Now, we have added the BrdU images of scramble and shRalB cells in the supplemental figure S9 and cited this information in the results section 2.3.

  1. In paragraph 2.4. the authors say that the expression of p16, p53, p21 and Rb1 is different in GBM6 and GBM65 cell lines. What does this different expression depend on?

      Brennan et al. 2013 (the Somatic Genomic Landscape of Glioblastoma, Cell 2013, 155, 462-477) showed that there are three signaling pathways frequently altered in most of the GBM, Ras, p53 and Rb1. p53 regulates senescence and apoptosis signaling and appears inactivated in 23% of glioblastoma. Rb1 pathway controls cell cycle and the most frequent deletions in this pathway are p16 (61%) and Rb1 (7.6%).

Now, we have commented this issue in discussion section: “In agreement with our data, Brennan et al, 2013 showed that GBM tumors frequently bear mutation in the TP53 and CDKN2A/p16 loci and less frequently in the RB1 locus [43]……..We consider that the differences observed in our work (also for p21 and 27) reflect the genetic variability among GBM tumors, and difficult to find a common pathway involved in senescence-like phenotype after RalB downregulation in GBM.”

Reviewer 2 Report

In the manuscript “Antitumor effects of Ral-GTPases downregulation in glioblastoma” by Cemeli, T. et al., showed that targeted inhibition of Ral GTPase prevents the proliferation of primary glioblastoma cells by inducing senescence like response.  Authors have further demonstrated that downregulation of RalA and RalB reduces the viability of GBM cells in vitro and retards the growth of xenografted tumors in mice.  Finally the authors conclude that Ral-GTPases may be good therapeutic targets to retard tumors growth.

Figures 1A & 1B:  Why did authors selected samples 41 and 65 for pulldown assay?   U87MG is one of the most widely used GBM cell line.  Why did authors selected U251 cell line in their study?

Figure 2B:  Why combination of RelA and RelB knockdown was not carried out in GBM41

In materials and methods, the authors have briefly described about the procedure for primary GBM cell cultures.  How did authors characterize the developed primary cultures?  It would be better if the authors can provide data pertaining to the characterization of primary GBM cultures

Authors would have shown the knockdown in tumors harvested from the mice.  Moreover, since these cells are programmed (by knocking down the RalB) it is expected that the tumor development will not take place.  Authors are requested to demonstrate the senescence induction even in xenografted tumors.  In addition, mechanisms such as proliferation (Ki67 staining), angiogenesis (CD31 staining) and apoptosis (TUNEL staining), which control tumor development should have been demonstrated by the authors.

It would have been better if authors could demonstrate the tumor growth inhibition by using a pharmacological agent (if available) rather than using a permanent knockdown approach. 

Author Response

Comments and Suggestions for Authors R2

In the manuscript “Antitumor effects of Ral-GTPases downregulation in glioblastoma” by Cemeli, T. et al., showed that targeted inhibition of Ral GTPase prevents the proliferation of primary glioblastoma cells by inducing senescence like response.  Authors have further demonstrated that downregulation of RalA and RalB reduces the viability of GBM cells in vitro and retards the growth of xenografted tumors in mice.  Finally the authors conclude that Ral-GTPases may be good therapeutic targets to retard tumors growth.

Figures 1A & 1B:  Why did authors selected samples 41 and 65 for pulldown assay?   U87MG is one of the most widely used GBM cell line.  Why did authors selected U251 cell line in their study?

A priori, we assumed that the accumulation of RalB in GBM also involved an increase in Ral GTPase activity. To check this possibility, we tested the pull down of RalB GTP with a couple of samples with different levels of RalB such as GBM65 and GBM41 (for the GBM6 we had no culture available at that time). Although the correlation with active forms is more difficult, the data would indicate that there is an increase in active forms or at least that all the RalB that accumulates is not in the inactive form.

In preliminary works, we have observed a better range of downregulation of RalA and RalB by RNA interference in U251MG cell line compared with other GBM cell lines such as U87MG. In addition, we have observed a RalB enrichment level in U251MG similar to the primary GBM (Figure 1A).

Figure 2B:  Why combination of RelA and RelB knockdown was not carried out in GBM41

The establishment of the primary cultures was carried out immediately after receiving the biopsy sample and, after few passages, the yield in cell number was variable among the different samples. As cited above, from the 10 primary samples of GBM (Figure 1), GBM6 and GBM65 are among the first we established in culture. Later, we also established GBM41. In the first phase of experiments, in the case of GBM41, the availability of cells was lower and we considered more important do not increase the number of passages. As the results of GBM6 and GBM65 showed that the double knockdown (RalA RalB) was not different from the single knockdowns, we assumed the results obtained only with the single knockdowns to be adequate.

In materials and methods, the authors have briefly described about the procedure for primary GBM cell cultures. How did authors characterize the developed primary cultures?  It would be better if the authors can provide data pertaining to the characterization of primary GBM cultures.

In previous experiments, after establishment of the primary culture, cell lines were analyzed by immunofluorescence for specific glioblastoma markers such as GFAP and IDH1R132. Primary cell lines retained the expression of Glial fibrillary acidic protein (GFAP) as reliable marker of astrocytic or glial cells (Nàger et al, ref 53), and were IDH wild type (Cemeli et al, ref 24).

The methylation status of the MGMT promoter was resolved in primary GBM samples and cell lines by pyrosequencing after bisulfite treatment. The expression of MGMT was determined in GMB6 and GBM65 cells by immunoblot (Cemeli et al, ref 24). Now, these data is mentioned in M&M section 4.1.

Authors would have shown the knockdown in tumors harvested from the mice.  Moreover, since these cells are programmed (by knocking down the RalB) it is expected that the tumor development will not take place.  Authors are requested to demonstrate the senescence induction even in xenografted tumors.  In addition, mechanisms such as proliferation (Ki67 staining), angiogenesis (CD31 staining) and apoptosis (TUNEL staining), which control tumor development should have been demonstrated by the authors.

Regarding senescence, xenografted tumors were fixed and included in paraffin, and we have not been able to determine in these conditions staining with SC-B-Gal. For apoptosis, we have not detected the presence of cleaved caspase 3 (Cleaved Caspase-3 (Asp175) (5A1E) Rabbit mAb #9664). However, with this negative result, we cannot conclude a total absence of apoptosis.  

Now, we have determined proliferation by Ki67 staining (Supplemental Figure S10). Subcutaneous tumors from RalB-downregulated cells showed a significant reduction of proliferation as expected. In our samples, the CD31 staining was barely detected and we could not determine relevance in angiogenesis. We have commented ki67 data in results section 2.5 and in M&M section 4.8.

It would have been better if authors could demonstrate the tumor growth inhibition by using a pharmacological agent (if available) rather than using a permanent knockdown approach. 

That was the original idea. Theodorescu’s lab (Yan et al, Nature. 2014 Nov 20; 515(7527):443-7) described the Ral inhibitors BQU57 and RBC8. These compounds inhibit tumour xenograft growth to a similar extent to the depletion of Ral using RNA interference. We have been unable to confirm the efficiency of Ral-inhibitors BQU57 and RBC8 in our experiments. Even though inhibitors from different sources and designs were tested, the results were not reproducible. I contacted Dan Theodorescu who told me about the existence of stability problems of the compounds (personal communication). We have cited this constrain in the discussion section.

Round 2

Reviewer 1 Report

All of my requests were satisfied by the Authors. In my opinion the work could be accepted in the present form.

Reviewer 2 Report

Authors have addressed the queries to the satisfaction, hence, the current version of the manuscript may be considered for publication